# Retrieval-Augmented Generation with Conflicting Evidence

**Han Wang    Archiki Prasad    Elias Stengel-Eskin    Mohit Bansal**

University of North Carolina at Chapel Hill
{hwang, archiki, esteng, mbansal}@cs.unc.edu

## Abstract

Large language model (LLM) agents are increasingly employing retrieval-augmented generation (RAG) to improve the factuality of their responses. However, in practice, these systems often need to handle ambiguous user queries and potentially conflicting information from multiple sources while also suppressing inaccurate information from noisy or irrelevant documents. Prior work has generally studied and addressed these challenges in isolation, considering only one aspect at a time, such as handling ambiguity or robustness to noise and misinformation. We instead consider multiple factors simultaneously, proposing (i) RAMDocs (Retrieval with Ambiguity and Misinformation in Documents), a new dataset that simulates complex and realistic scenarios for conflicting evidence for a user query, including ambiguity, misinformation, and noise; and (ii) MADAM-RAG, a multi-agent approach in which LLM agents debate over the merits of an answer over multiple rounds, allowing an aggregator to collate responses corresponding to disambiguated entities while discarding misinformation and noise, thereby handling diverse sources of conflict jointly. We demonstrate the effectiveness of MADAM-RAG using both closed and open-source models on AmbigDocs – which requires presenting all valid answers for ambiguous queries – improving over strong RAG baselines by up to 11.40%, and on FaithEval – which requires suppressing misinformation – where we improve by up to 15.80% (absolute) with Llama3.3-70B-Instruct. Furthermore, we find that our proposed RAMDocs dataset poses a challenge for existing RAG baselines (the most performant Llama3.3-70B-Instruct only yields up to a 32.60 exact match score), as it requires handling conflicting information due to ambiguity, noise, and misinformation *simultaneously*. While MADAM-RAG begins to address these conflicting factors, our analysis indicates that a substantial gap remains, especially when increasing the level of imbalance in supporting evidence and misinformation.[1]

## 1 Introduction

Retrieval-augmented generation (RAG) enables large language models (LLMs) to generate more accurate and reliable responses by incorporating retrieved external information (Lewis et al., 2020; Guu et al., 2020; Wang et al., 2021), mitigating issues such as hallucination (Zhang et al., 2023) and outdated parametric knowledge (Kasai et al., 2023). Indeed, recent LLM interfaces and AI-powered search engines, such as ChatGPT (OpenAI, 2024), Claude (Anthropic, 2025), Google Search with AI Overviews (Google, 2024b), and Microsoft Bing Chat (Microsoft, 2024), have integrated retrieval capabilities that enable them to access vast amounts of information from the internet, allowing them to summarize search results and provide more accurate and up-to-date answers. This kind of RAG also features prominently in "deep research" techniques that frame search as an agent-driven process in which a research agent collects and summarizes online sources (Google, 2024a; OpenAI, 2025). However, a core challenge faced by all of these approaches is that information retrieved from the internet can be *conflicting, noisy, and unreliable* – retrieved documents might contain

---

[1]Our data and code is publicly available at: https://github.com/HanNight/RAMDocs.

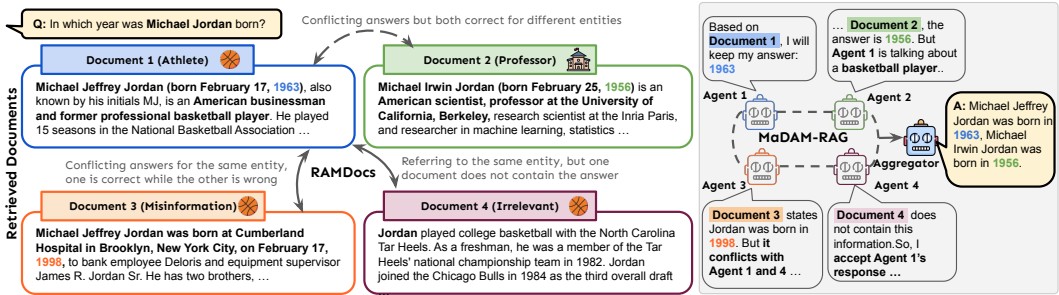

Figure 1: An example from our RAMDocs dataset (left) illustrating multiple sources of conflict in retrieved documents. Conflict may occur because of ambiguity in the query (in this case, different referents for Michael Jordan), but also because of misinformation from incorrect documents and noise from irrelevant ones. MADAM-RAG (right) addresses this through multi-agent debate, where each agent summarizes and represents the information in one document. Agents discuss their responses across multiple rounds, with the final answers being combined via an aggregator module that summarizes the discussion.

misinformation, unverified claims, and AI-generated content that may be inaccurate or misleading (Pan et al., 2023b; Augenstein et al., 2023). Moreover, queries themselves may be *ambiguous and underspecified*, leading to the retrieval of conflicting yet factually accurate information from different sources (Wan et al., 2024; Lee et al., 2024).

In practice, different kinds of conflict will occur within the same system, each with a different expected behavior. For instance, conflict due to an ambiguous query ought to be treated differently from conflict due to noisy or incorrect documents. As illustrated in Figure 1, a model should present *multiple valid answers for an ambiguous query* (i.e., conflict in documents 1 and 2), but should *filter out misinformation and noise* (i.e., the conflict introduced by documents 3 and 4). Existing datasets generally capture only one aspect of conflict in retrieved documents: e.g., AmbigDocs (Lee et al., 2024) focuses on ambiguous queries, while FaithEval (Ming et al., 2024) assesses an LLM's ability to handle conflict due to misinformation or noise. Similarly, approaches to improve robustness of RAG systems focus primarily on improving the retrieved content by filtering out incorrect, irrelevant or even malicious information, e.g., due to conflicts with parametric knowledge (Wang et al., 2024a; Jin et al., 2024) or poisoning attacks (Weller et al., 2024; Zhou et al., 2025). However, approaches designed to choose only one correct answer may fail to generalize to settings like ambiguity, where conflict is expected and multiple valid answers exist, e.g., when there is legitimate uncertainty in the retrieved documents or ambiguity in the query (Lee et al., 2024; Wan et al., 2024). Therefore, as illustrated in Figure 1, RAG systems need to balance the *tradeoff* between presenting conflicting information (to disambiguate a user query with multiple valid answers) while also selectively filtering out misinformation and noise.

To address this tradeoff, we introduce Multi-agent Debate for Ambiguity and Misinformation in RAG (MADAM-RAG), a *unified multi-agent approach* designed to handle multiple diverse cases of information conflict, with different expected behaviors depending on the source of conflict. In case of ambiguous queries, MADAM-RAG can opt to show multiple responses while also removing misinformation or noise. While prior work processes and filters retrieved documents collectively (Wang et al., 2024a; Weller et al., 2024), our approach assigns each document to an independent agent (instantiated from the same LLM), which generates an intermediate response based solely on its input document. Next, these multiple agents debate over the merits and evidence backing their intermediate responses, iteratively updating their answers across a dialogue. For instance, in Figure 1 (right), the multi-agent interaction reveals that documents 1 and 2 refer to two different people, i.e., a basketball player and a professor, thus rightfully corresponding to two different answers, one for each disambiguated entity. At the same time, the debate can uncover that documents 3 and 4 also refer to the same basketball player and are unreliable when compared to document 1. Finally, the discussion is summarized by an aggregator module that synthesizes a coherent final response from the agent discussions (cf. Figure 1).

In addition to evaluating MADAM-RAG against strong RAG baselines on existing datasets like FaithEval (Ming et al., 2024) and AmbigDocs (Lee et al., 2024) that measure one source of conflict (i.e., due to misinformation or due to ambiguity, etc.), we introduce RAMDocs, a unified dataset that combines multiple sources of conflict corresponding to a single query, covering ambiguity, noise from unrelated documents, and misinformation. Built on top of AmbigDocs, RAMDocs retains disambiguated queries and answers, and augments them with misinformation documents (created by replacing correct entities with plausible but incorrect ones) and noisy documents (retrieved passages that are topically irrelevant to the query) which is likely to occur in real-world retrieval settings. Unlike prior datasets (Lee et al., 2024; Wan et al., 2024) that assume a uniform distribution of supporting documents, we also introduce cases where different answers have uneven document support, testing how model outputs change across varying ratios of representation for each perspective.

Empirically, across three LLMs: Llama3.3-70B-Inst (Llama Team, 2024), Qwen2.5-72B-Inst (Qwen et al., 2025) and GPT-4o-mini (Hurst et al., 2024), we show that MADAM-RAG outperforms Astute-RAG (Wang et al., 2024a), which iteratively clusters and combines retrieved documents while filtering misinformation, by 11.40% (absolute accuracy) on AmbigDocs (measuring the model's ability to handle ambiguous queries) with Llama3.3-70B-Inst and by 13.10% on FaithEval (measuring robustness to misinformation) when using Qwen2.5-72B-Inst. Furthermore, when compared to relying solely on the LLM's parametric knowledge and the standard RAG pipeline which concatenates all retrieved documents into the model's context, we demonstrate that MADAM-RAG beats the parametric knowledge baseline by 7.50% on FaithEval, and the concatenated prompt baseline by 11.50% on AmbigDocs with GPT-4o-mini. Additionally, our ablations with Llama3.3-70B-Inst verify the salience of MADAM-RAG's aggregator and multi-round discussion, with accuracy improvements of 19% and 5.30% on FaithEval. Lastly, we demonstrate the utility of RAMDocs, which tests on ambiguity, noise, and misinformation together, finding that all baselines degrade in performance as we increase the imbalance in supporting documents per answer. While MADAM-RAG helps mitigate these drops in performance, there remains large room for improvement on RAMDocs which we introduce as a challenge for future work.

## 2 Related Work

**Retrieval-Augmented Generation.** Retrieval-augmented generation (RAG) has emerged as a crucial technique for enhancing language models via integrating external knowledge retrieval instead of solely relying on parametric knowledge (Lewis et al., 2020; Karpukhin et al., 2020; Cheng et al., 2021). Variants such as REALM (Guu et al., 2020), Fusion-in-Decoder (FiD) (Izacard & Grave, 2021), and RETRO (Borgeaud et al., 2022) have further optimized retrieval methods and document encoding strategies to enhance efficiency and scalability. However, these do not fully address the critical challenge of handling conflicting information across multiple retrieved documents. Prior work, such as Chen et al. (2022); Zou et al. (2024), has shown that RAG systems often propagate misinformation if retrieved documents contain errors, or arbitrarily choose an answer or use parametric knowledge to break ties when documents provide conflicting claims. Unlike existing solutions like SELF-RAG (Asai et al., 2024) that use LLM-generated critiques, Astute RAG (Wang et al., 2024a) that uses the LLM's parametric knowledge and clustering to combine retrieved documents and filter outliers or misinformation, and Speculative RAG (Wang et al., 2025), which uses a small specialist LM to draft multiple answers in parallel from distinct subset documents, and a larger generalist LM to verify and select the best one, MADAM-RAG employs multi-agent debate over several rounds, wherein each agent gets the opportunity to revise its response as well as influence other agents in each round. Moreover, in contrast to Chang et al. (2024), who use a multi-agent framework solely for scoring and filtering out noisy documents, our multi-agent approach also handles other scenarios with conflicting evidence by suppressing misinformation and presenting multiple valid answers for ambiguous queries.

**RAG Evaluation Benchmarks.** Several benchmarks have been proposed to evaluate RAG systems under various conditions. AmbigNQ (Min et al., 2020) and AmbigDocs (Lee et al., 2024) assess the ability to handle questions with multiple valid answers, with the latter extending information about ambiguous entities across multiple documents, leading us to test on AmbigDocs. However, these datasets do not consider noise or misinformation. On

the other hand, the RGB dataset (Chen et al., 2024) introduces retrieval noise, testing how well models respond to partially relevant content in the retrieved documents. HAGRID (Kamalloo et al., 2023) focuses on attribution in generative QA, while CRAG (Yang et al., 2024) presents a comprehensive benchmark covering diverse question types across five domains, varying in entity popularity and temporal sensitivity. The DRUID dataset (Hagström et al., 2025) contains internet-retrieved evidence annotated for stance and relevance, including both misinformation and insufficient or irrelevant evidence. Note that each of these datasets assumes that a given question has a single correct answer and thus they do not address scenarios with multiple conflicting yet valid answers, which our work explicitly targets. We unify these separate lines of work by introducing RAMDocs that includes multiple valid answers for each query (due to ambiguity) as well as uneven numbers of supporting documents and conflicting evidence from misinformed and noisy documents.

**Knowledge Conflict in LLMs.**   Recent studies have highlighted the challenge of knowledge conflict in LLMs, where inconsistent or conflicting information may arise from either internal parametric memory or retrieved external context. A comprehensive survey by Xu et al. (2024b) categorizes knowledge conflicts into intra-context, inter-context, and parametric conflicts, and outlines the limitations of current methods in resolving them. While prior work has made strides in addressing parametric and single-document conflicts (Gao et al., 2023; Wang et al., 2024b; Huang et al., 2025), recent studies highlight the difficulty of resolving disagreements across multiple contexts or retrieved documents (Chen et al., 2022; Su et al., 2024). Such inter-context conflict can stem from misinformation (Pan et al., 2023a; Zhou et al., 2023) or outdated knowledge (Kasai et al., 2023), and has been shown to impair factual accuracy (Jin et al., 2024). Our MADAM-RAG approach addresses this gap by modeling each document by a separate LLM agent, using multi-agent debate to identify potential ambiguity and aggregating multiple valid answers while also suppressing and resolving conflicting evidence stemming from misinformation and noisy retrieval.

## 3   RAMDocs: Retrieval with Ambiguity & Misinformation in Documents

Measuring the performance of RAG systems in real-world settings requires assessing each system's ability to cope with inter-context conflict—disagreements among retrieved documents—which may arise due to ambiguity, misinformation, or noise. These conflict sources differ in nature: ambiguity involves different but valid answers from underspecified queries (Lee et al., 2024), misinformation refers to factually incorrect content (Ming et al., 2024), and noise includes irrelevant or weakly related documents. Moreover, different kinds of conflict demand different solutions: ambiguity should not be treated the same way as misinformation or noise; these solutions may at times conflict with each other. While existing datasets address these factors separately, in practice they will appear together and may interact in unexpected ways. This makes measuring multiple kinds of conflict in a single dataset crucial, and motivates the need for such a representative set of retrieved documents.

**Ambiguous Queries.**   To rigorously evaluate the robustness of current RAG systems, we present RAMDocs, a dataset designed to simulate real-world retrieval challenges, including conflicting information, noise, and misinformation. Building upon AmbigDocs (Lee et al., 2024) that focuses on evaluating ambiguity by presenting multiple valid answers per query, we extend it by introducing additional real-world retrieval complexities. Specifically, we randomly sample between 1 to 3 correct answers for each ambiguous query from the dataset, resulting in 500 queries. This setup ensures that models must not only recognize ambiguity but also operate under realistic constraints where only partial information may be retrieved.

**Distribution of Supporting Documents.**   Second, we vary the number of documents supporting each valid answer for a query, recognizing that real-world scenario where multiple answers are often backed by different numbers of sources. Using the Brave Search API[2], we retrieve multiple supporting documents per disambiguated query and split them into text chunks (up to 100 words each), selecting only those containing the correct answer as the supporting evidence. However, unlike traditional datasets where the number of supporting documents per answer is fixed (Lee et al., 2024; Wan et al., 2024), we also randomly vary the number of documents supporting each valid answer. Therefore,

---

[2]https://brave.com/search/api/

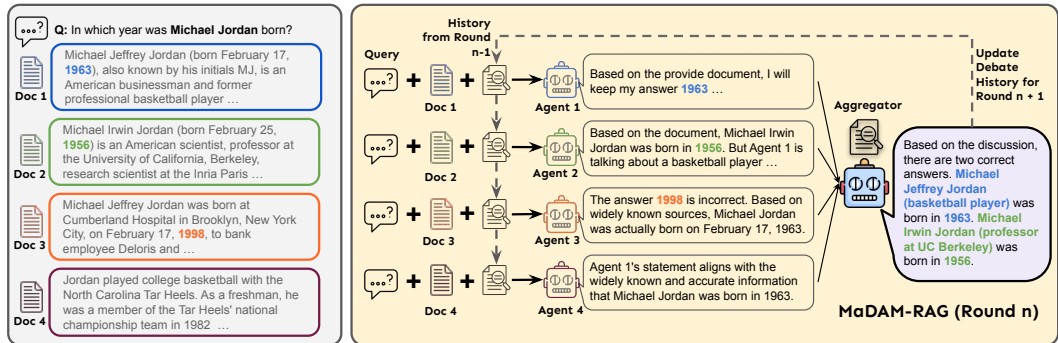

Figure 2: MADAM-RAG operates by assigning each document to a separate agent. Agents are responsible for summarizing and representing their documents, and engage in a multi-agent, multi-round debate with each other to filter out misinformation and noise and address ambiguity. An aggregator then summarizes the discussion into a single response.

each query in RAMDocs is associated with 1-3 disambiguated answers, and each answer is supported by 1-3 documents, resulting in an average of 2.20 valid answers and 3.84 supporting documents containing a valid answer per query. This design introduces realistic retrieval imbalances, where some valid answers are well-documented while others (despite being valid) may appear in only a few sources, forcing the model to identify ambiguity without being spuriously biased towards the most frequent answer.

**Misinformation and Noisy Documents.** Finally, for each query, we include documents containing misinformation and noise to evaluate an RAG system's resilience against unreliable content. To incorporate misinformation, we include an additional retrieved document and replace the correct answer with a misleading or incorrect entity, ensuring the misinformation blends naturally into the document context. This is similar to the approach in Longpre et al. (2021), who replace the original answer in the context with a similar type of answer, but here, we adapt the procedure to modify the retrieved documents. We randomly determine the number of misinformation documents added to each query from 0-2, reflecting the unpredictable nature of misinformation in real-world retrieval. Additionally, we introduce noisy documents for each query (sampled from [0, 2]), which are either irrelevant to the query or contain low-quality content, to assess the system's ability to filter out noise.

Overall, by introducing randomness in the number of correct answers, the number of supporting documents per answer, and adding misinformation and noisy documents for each query in RAMDocs, we design a challenging setting simulating the variability and unpredictability of real-world retrieval scenarios. The resulting dataset comprises 500 queries, with an average of 2.20 valid answers per query. For each query, RAMDocs comprises an average of 5.53 documents in the retrieved pool, out of which an average of 3.84 documents support a valid answer and the remaining 1.70 documents correspond to misinformation and noise. Empirically, we demonstrate that LLMs struggle in this challenging setting in Section 5.2 and expand on how performance degrades with each of these factors in Section 6. We refer readers to Appendix A for detailed dataset statistics and distribution plots.

## 4 MADAM-RAG: Multi-agent Debate for Ambiguity and Misinformation in RAG

We propose MADAM-RAG, a structured, multi-agent framework designed to handle inter-document conflicts, misinformation, and noise in retrieved content (illustrated in Figure 2). Our overall framework consists of three key components: (i) independent LLM dialogue agents that generate intermediate responses conditioned on a single document, (ii) a centralized aggregator, and (iii) an iterative multi-round debate process. Let $q$ denote the user query, and let $D = \{d_1, d_2, \ldots, d_n\}$ be the set of retrieved documents.

**Role of a Single LLM Agent.** Each document $d_i \in D$ is assigned to an agent $\mathcal{L}_i$, which processes the query $q$ and it corresponding document $d_i$. The agent generates an intermediate response $r_i = \mathcal{L}_i(q, d_i)$ based on the query and the assigned document *independently*, i.e.,

without the influence of other documents. This design has two key features: (i) each agent has the chance to thoroughly review all the information in a given document, reducing chances of overlooking details in case of very long contexts (Liu et al., 2024); and (ii) prevents the agent's response to be unduly impacted by document frequency. For instance, in Figure 2 only one document references the professor "Michael I. Jordan" which can be missed if all documents are considered collectively due to long context, frequency and position bias (Zhao et al., 2021; Liu et al., 2024). Furthermore, this design ensures that each retrieved document contributes its perspective with minimal interference from other documents – this separation is critical in settings where sources may present conflicting yet individually valid information. In a single round, each agent $\mathcal{L}_i$ produces it intermediate response $r_i$, forming a set of intermediate outputs $\mathcal{R}^{(t)} = \{r_i^{(t)}\}_{i=1}^n$, where $t$ denotes current debate round. While the initial round relies solely on the document and query, in later rounds each agent also receives a summary of the prior round's responses (an aggregator-generated summary described below), allowing for informed revisions during multi-round debate.

**Aggregator Module.** The aggregator $\mathcal{A}$ receives the outputs from all agents in round $t$, and generates a summarized answer with explanation $(y^{(t)}, e^{(t)}) = \mathcal{A}(\mathcal{R}^{(t)})$. To mitigate position bias, we shuffle agent responses before passing them to the aggregator. The aggregator considers different agents' responses, resolves inconsistencies, and synthesizes reliable answers based on available evidence, taking a global view over the entire conversation which allows it to take into account the source of a conflict and whether it is valid. In cases of ambiguous queries such as the example in Figure 2, where two answers refer to different individuals named "Michael Jordan"—the aggregator recognizes that both 1963 and 1956 are plausible birth years corresponding to distinct entities (the basketball player and the professor). In contrast, it flags 1998 (from the misinformation document) as factually inconsistent and irrelevant.

**Multi-round Debate.** To effectively handle conflicting retrieved evidence, we conduct multi-round debate among the agents; in each round $t$, agents are given the aggregated answer and explanation generated by the aggregator from the previous round, and they are prompted to reflect and optionally revise their own answers: $r_i^{(t)} = \mathcal{L}_i(q, d_i, y^{(t-1)}, e^{(t-1)})$. This iterative process allows agents to defend, challenge, or revise their claims, promoting convergence toward a consistent, evidence-supported output. As illustrated in Figure 2, some agents initially propose different birth years for "Michael Jordan" based on distinct individuals (1963 for the basketball player, 1956 for the professor). In subsequent rounds, agents 1 and 2 recognize that both answers are valid due to ambiguity, and thus retain their positions without being forced to collapse toward a single answer. In contrast, agent 3, supporting the fabricated year 1998 – stemming from misinformation in document 3 – fails to justify its position when challenged, and the incorrect answer is disregarded. This back-and-forth encourages selectivity in the presence of factual inconsistency, while preserving diversity in the case of legitimate ambiguity. The debate converges as unsupported claims are filtered out and well-justified answers remain.

**End of Debate and Final Answer.** The debate proceeds for a fixed maximum number of rounds $T$. To improve efficiency and avoid unnecessary computation, we incorporate an early stopping criterion. Specifically, if all agents retain their answers from the previous round, that is, $\forall i, r_i^{(t)} = r_i^{(t-1)}$, we consider the debate to have converged at round $t_{\text{end}} \in [1, T]$ and the final answer is determined $y = \mathcal{A}(\mathcal{R}^{(t_{\text{end}})})$. This iterative process ensures that misleading or unsupported claims are scrutinized, enhancing the overall robustness of the final response. By structuring information synthesis in this manner, MADAM-RAG improves reliability, mitigates the influence of misinformation, and enhances decision-making in conflicting retrieval scenarios, making it a more effective solution for real-world applications of RAG systems.

# 5 Experiments and Results

## 5.1 Experimental Setup

**Datasets.** We evaluate MADAM-RAG on three datasets: FaithEval's inconsistent subset (Ming et al., 2024), AmbigDocs (Lee et al., 2024), and our proposed RAMDocs. FaithEval

tests whether LLM-generated responses remain faithful to retrieved evidence, particularly in the presence of subtle factual perturbations (i.e., misinformation) over 1000 instances. AmbigDocs focuses on evaluating ambiguity in multi-document settings by pairing each question with multiple documents that may support different valid answers. For Ambig-Docs, we sample 1000 instances from its test set. Our RAMDocs is constructed to simulate real-world retrieval conditions, including multiple valid answers with varied document support, inter-document conflicts, misinformation, and noise over 500 unique queries, obtained from AmbigDocs. These datasets together enable a comprehensive assessment of model robustness under ambiguous, conflicting, and potentially misleading retrieval conditions.

**Metrics.** We evaluate the outputs using *exact match* under a strict criterion: an output is considered correct if and only if it includes *all* gold answers and *no* incorrect or misleading answers supported only by misinformation documents. This ensures that the model is both comprehensive and factually precise in its generation.

**Models.** We test on two open-source post-trained language models: Llama3.3-70B-Instruct (Llama Team, 2024) and Qwen2.5-72B-Instruct (Qwen et al., 2025), as well as GPT-4o-mini (Hurst et al., 2024), representing a closed-source frontier model.

**Baselines.** For each LLM, we compare MADAM-RAG with the aforementioned LLMs (for $T = 3$ maximum rounds) against the following baselines (refer to prompts in Appendix E):

- **No RAG**: Prompt the LLM with the question only, without any retrieved documents, to elicit its answer purely on internal parametric knowledge.
- **Concatenated-prompt**: A standard RAG pipeline where all retrieved documents are concatenated and passed as input to the model along with the query. This is used to test the LLM's ability to jointly reason over the full set of retrieved documents – identifying ambiguity, detecting misinformation, and resolving conflicts in a single inference – taking advantage of its long context window.
- **Single Agent with Self-reflection**: A three-step prompting strategy adapted from Madaan et al. (2023); Huang et al. (2024): 1) prompt LLM to generate an initial answer; 2) prompt LLM to review its previous answer and produce feedback; 3) prompt LLM to answer the question again with the feedback. We conduct two rounds of self-reflection.
- **Self-RAG** (Asai et al., 2024): Self-RAG instruction-tunes an LLM to generate self-reflection tags that guide dynamic retrieval and critique the relevance of retrieved documents before answering. It processes multiple documents in parallel, producing separate responses for each. Since our setting allows multiple correct answers, we forward all responses with critique scores to the aggregator to generate the final answer, rather than selecting the highest-scoring one as in the original method.
- **Speculative RAG** (Wang et al., 2025): A framework that uses a smaller, distilled specialist LM to generate drafts in parallel that are then fed to a larger generalist LM to verify and select the best draft. Each draft is generated from a distinct subset of retrieved documents, offering diverse perspectives on the evidence. See details in Appendix B.
- **Astute RAG** (Wang et al., 2024a): A recent method that aligns retrieved content with the model's parametric knowledge to resolve inconsistencies. The framework generates internal parametric knowledge document, clusters retrieved and parametric knowledge documents into consistent or conflicting groups, and then selects the most reliable cluster to generate the final answer. See details in Appendix C.

### 5.2 Main Results

**MADAM-RAG outperforms baselines across tasks.** As shown in Table 1, MADAM-RAG consistently outperforms concatenated-prompt and Astute RAG baselines across all three datasets and model backbones. The gains are particularly notable on AmbigDocs and RAMDocs, where ambiguity and conflicting evidence require structured resolution; for instance, MADAM-RAG outperforms Astute RAG by 11.40% with Llama3.3-70B-Inst and by 12.90% with Qwen2.5-72B-Inst. On FaithEval, a dataset with conflicting evidence but only a single correct answer, MADAM-RAG yields the highest performance, surpassing concatenated-prompt baseline by 15.80%, and 19.20% with Llama3.3-70B, and Qwen2.5-72B models, respectively, demonstrating its robustness to subtle misinformation. Compared to single-agent baseline with self-reflection baseline, MADAM-RAG achieves gains of 17.40% on FaithEval, 3.70% on AmbigDocs, and 4.00% on RAMDocs with Llama3.3-70B-Inst. These

| Model | Method | FaithEval | AmbigDocs | RAMDocs |
|---|---|---|---|---|
| SelfRAG-Llama2-13B | Self-RAG | 12.40 | 55.00 | 26.80 |
| Llama3.3-70B-Inst | No RAG | 26.70 | 4.30 | 5.80 |
| | Prompt-based | 27.30 | 54.20 | 32.60 |
| | Single Agent with Self-reflection | 25.70 | 54.50 | 30.40 |
| | Speculative RAG | 41.80 | 44.30 | 30.60 |
| | Astute RAG | 37.10 | 46.80 | 31.80 |
| | MADAM-RAG | **43.10** | **58.20** | **34.40** |
| Qwen2.5-72B-Inst | No RAG | 26.40 | 1.80 | 4.20 |
| | Prompt-based | 38.50 | 41.20 | 20.60 |
| | Single Agent with Self-reflection | 39.40 | 29.40 | 21.80 |
| | Speculative RAG | 56.20 | 13.40 | 22.20 |
| | Astute RAG | 44.60 | 39.80 | 20.80 |
| | MADAM-RAG | **57.70** | **52.70** | **26.40** |
| GPT-4o-mini | No RAG | 31.00 | 1.00 | 2.50 |
| | Prompt-based | 21.00 | 51.50 | 25.00 |
| | Single Agent with Self-reflection | 36.00 | 44.50 | 22.50 |
| | Speculative RAG | 36.50 | 22.50 | 23.00 |
| | Astute RAG | 34.00 | 15.00 | 13.00 |
| | MADAM-RAG | **38.50** | **63.00** | **28.00** |

Table 1: MADAM-RAG outperforms baselines across datasets on different language models.

improvements show some of the shortcomings of single-agent systems: while single-agent approaches like self-reflection (Madaan et al., 2023; Huang et al., 2024) can identify internal inconsistencies, they still process concatenated input and are therefore vulnerable to context length limits, frequency bias, and order effects (Liu et al., 2024; Zhao et al., 2021). Notably, in Table 1, we find that solely using the parametric knowledge of the LLM (i.e., the No RAG baseline which does not see any of the retrieved documents) is somewhat effective on FaithEval owing to the fact that it *does not* have access to the misinformation in documents. However, relying on parametric knowledge alone is not sufficient as we find that LLMs when presented with ambiguous queries have the propensity to only present one valid answer, thereby, leading to poor performance on AmbigDocs and RAMDocs. On the other hand, we demonstrate that MADAM-RAG has the ability to ignore misinformation in retrieved documents while effectively utilizing evidence supporting multiple valid answers.

**RAMDocs is a challenging RAG setting.** Across all models and baselines (including MADAM-RAG), Table 1 shows that our dataset, RAMDocs, is significantly more challenging. Across the board, systems generally perform worse on RAMDocs than on FaithEval or AmbigDocs; this holds even when systems obtain non-trivial accuracy on both settings, e.g., MADAM-RAG obtains 57.70 and 52.70 accuracy on FaithEval and AmbigDocs for Qwen2.5-72B-Inst but only 26.40 for RAMDocs. Baseline methods see similarly low scores on RAMDocs: using Astute RAG with Llama3.3-70B-Inst yields a score of 31.80. MADAM-RAG does increase performance on RAMDocs: when using MADAM-RAG, the performance increases to 34.40. The same trend is persistent on a frontier LLM (GPT-4o-mini), where the performance peaks at 28.00 with MADAM-RAG. Overall, these results indicate that jointly handling ambiguity, misinformation, and noise in retrieved documents remains an ongoing challenge for LLMs – a situation that is likely to be encountered by information-seeking agents on the Internet. While the gains from MADAM-RAG at this challenging setting are encouraging, RAMDocs leaves room for improvement for future work to build upon.

## 6 Ablations and Analysis

### 6.1 Importance of Using the Aggregator and Multiple Rounds of Debate

**Setup.** To isolate the contributions of the aggregator and the multi-round debate mechanism in MADAM-RAG, we conduct controlled ablations along two axes: (i) enabling or disabling the aggregator, and (ii) varying the number of debate rounds (1, 2, or 3) on FaithEval and RAMDocs datasets. We evaluate performance using four metrics: (i) accuracy (Acc.) requiring exact match with all correct and no incorrect answers (described in Section 5.1);

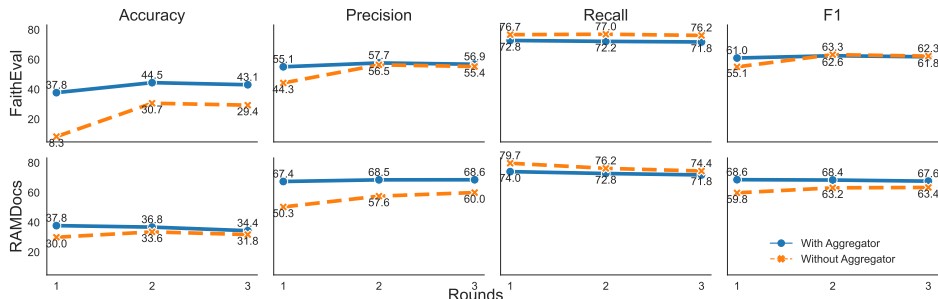

Figure 3: Ablation study on the importance of aggregator and multiple rounds of discussion.

(ii) Precision (Prec.), which measures the proportion of predicted answers that are correct; (iii) Recall (Rec.), which measures the proportion of correct answers that are successfully included in the model's response; and (iv) the F1 score that captures the overall coverage of correct and incorrect answers. For each setting, we use Llama3.3-70B-Instruct as the backbone model. In the setting without an aggregator, all agent responses are concatenated to produce a naive joint answer, whereas with an aggregator, MADAM-RAG generates a single response by explicitly comparing and reconciling rationales from each agent. The number of debate rounds controls the maximum number of times individual agents can potentially revise their answers based on previous history of discussion (cf. Figure 2).

**Takeaway.** As shown in Figure 3, both increasing the number of debate rounds and enabling the aggregator contribute to substantial gains. Without the aggregator, performance improves substantially with more rounds (e.g., +21.10 gain in accuracy from round 1 to 3 on FaithEval and +3.62 increase in F1 on RAMDocs), indicating that iterating allows agents to refine their answers and reduce error. Adding the aggregator provides a stronger boost, especially in earlier rounds: in round 1, accuracy improves from 30.00 to 37.80 and F1 from 59.79 to 68.63 on RAMDocs, reflecting the aggregator's ability to synthesize evidence and suppress misinformation effectively. Across rounds, the aggregator yields higher precision (e.g., 67.41 → 68.52 → 68.57 for RAMDocs), while recall tends to decrease slightly, suggesting a more selective and cautious answering strategy. Importantly, we argue that precision is more critical in conflict-heavy settings like RAMDocs as false positives (i.e., including misinformation) could be more harmful than false negatives (e.g., missing a correct answer). In such cases, it may be preferable for a model to be cautious and omit uncertain answers rather than risking introducing incorrect ones. The aggregator helps enforce this precision-reliability trade-off by aligning agent views early and mitigating the influence of conflicting or noisy sources. Overall, these results support the aggregator's role in improving answer faithfulness and trustworthiness, especially under realistic, imperfect retrieval conditions.

## 6.2 Impact of varying the number of Supporting Documents

**Setup.** To analyze how MADAM-RAG handles imbalance in evidence (i.e., number of supporting documents) across correct answers, we construct a controlled subset of 200 examples from RAMDocs, where two correct answers are supported by differing numbers of supporting documents. This simulates realistic retrieval imbalance where some valid answers may be underrepresented. For each query, we fix the number of documents supporting one correct answer to 1, and then vary the number of supporting documents for the other answer from 1 to 3. To avoid other confounding factors, we ensure all documents are factual and relevant, i.e., no misinformation or noise, and use Llama3.3-70B-Instruct.

**Takeaway.** From Figure 4(a), we observe that as the number of supporting documents increases, the performance of the two baselines decreases, falling by up to 8% when using concatenated-prompt. We explain this by noting that as the imbalance in the underlying evidence increases, the baselines have a greater propensity of the baselines to favor the answer with more supporting documents, and suppressing the underrepresented valid answer altogether increases. This is consistent with Stengel-Eskin et al. (2024)'s findings on parsing ambiguous sentences with mixed evidence, which also found that LLMs tend to follow the more attested interpretation. Furthermore, in Figure 4, we find that MADAM-RAG curtails this drop in performance as the level of imbalance increases, with an average

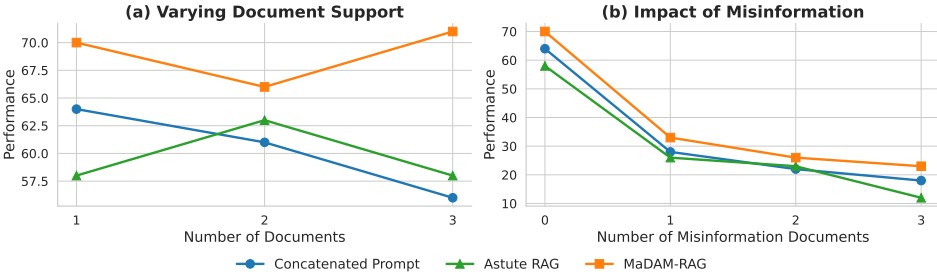

Figure 4: Performance of different methods with Llama3.3-70B-Instruct under setting (a) imbalance in supporting documents and (b) increasing the level of misinformation.

variance of 3.33 across rounds. This in turn demonstrates the ability of a single agent to champion a valid answer in the multi-agent discussion with minimal interference from other agents and the imbalance in the number of documents.

### 6.3  Impact of Increasing Misinformation

**Setup.**  Next, we evaluate how increasing the level of misinformation impacts performance of various RAG systems. Using RAMDocs, we curate a set of 200 queries where two correct answers each have one supporting document. We then incrementally introduce 1 to 3 documents that promote a factually incorrect alternative. The misinformation documents are created using the entity-swap strategy from FaithEval – replacing correct entities with plausible but incorrect alternatives while preserving fluency and context. We test how each method degrades under this pressure, using EM to measure whether the system selects only the correct answers and rejects the misinformation. This setting is particularly challenging as it simulates real-world retrieval pipelines that include inaccurate or adversarial content.

**Takeaway.**  Results in Figure 4(b) show that increasing the level of misinformation in retrieved evidence negatively impacts overall performance. For instance, both concatenated-prompt and Astute RAG baselines suffer a decrease of 46%, each. While MADAM-RAG also drops in performance, given a fixed degree of misinformation, MADAM-RAG yields the highest performance. This robustness stems in part from how evidence is processed: processing all documents jointly may obscure low-frequency but valid evidence, especially when misinformation or noise dominates. Our multi-agent setup encourages each document to be independently summarized and defended, allowing underrepresented but correct views to be preserved and strengthened through debate. We note that the performance degradation with more misinformation is to be expected, as higher degree of conflicting evidence can build mistrust around the query, making it harder for the LLM to distinguish factual documents from inaccurate ones. These findings align with those of Xu et al. (2024a), who find that even strong LLMs are susceptible to being persuaded of misinformation.

## 7  Conclusion

Building on the observation that real-world RAG agents will have to cope with conflict from a variety of sources, we construct the RAMDocs dataset, which includes conflict between documents stemming from a number of sources and with different expected behaviors. RAMDocs covers conflict due to query ambiguity, where it expects agents to return multiple correct answers, as well as conflict due to misinformation or noise, where it expects agents to return only correct answers. Moreover, RAMDocs varies the number of documents supporting each view, reflecting imbalances that may occur in retrieved data. To tackle this unique challenge, we also introduce MADAM-RAG, a multi-agent approach for handling conflicting evidence wherein each agent is assigned a separate document and agents debate with each other. The debate is then synthesized into an answer by an aggregator model, leading to improved performance across both standard RAG datasets as well as RAMDocs. Our analysis indicates the importance of the aggregator in MADAM-RAG and finds MADAM-RAG best handles scenarios of high conflict as we vary the amount of evidence for a given answer and varying amounts of misinformation. Taken together, these results point to future directions for improving on the challenging task posed by RAMDocs.

## Acknowledgments

We would like to thank the anonymous reviewers for their feedback. This work was supported by NSF-CAREER Award 1846185, DARPA ECOLE Program No. HR00112390060, the Microsoft Accelerate Foundation Models Research (AFMR) grant program, and NSF-AI Engage Institute DRL-2112635. Any opinions, findings, and conclusions or recommendations in this work are those of the author(s) and do not necessarily reflect the views of the sponsors.

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

# A RAMDocs: Dataset Statistics

To better understand the composition of our constructed dataset, we present summary statistics across key dimensions, including the number of correct and incorrect answers per example, the total number of documents retrieved, and the distribution of documents that support correct answers, incorrect answers, or contain irrelevant noise. These statistics reflect the degree of ambiguity, evidence imbalance, and misinformation within the dataset, providing insight into the challenges it poses for retrieval-augmented generation systems.

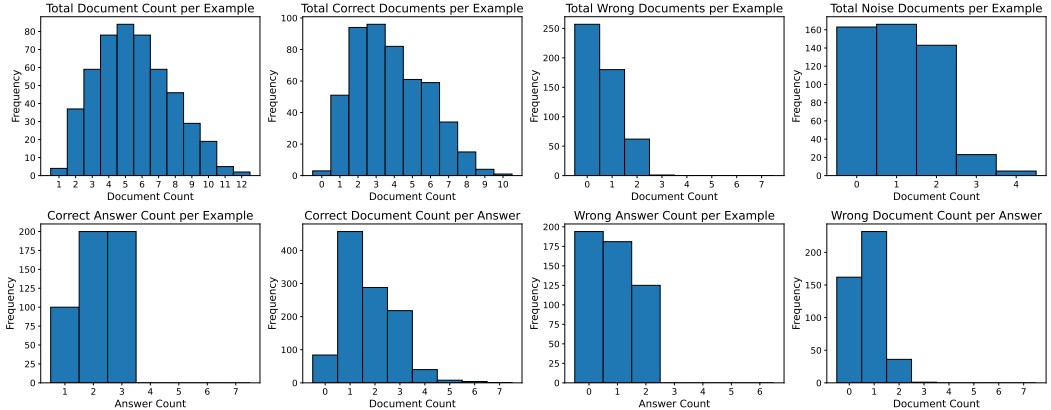

Figure 5: Dataset statistics across eight dimensions. The first row shows document-level properties per example: total number of documents (avg: 5.53), number of documents supporting correct answers (avg: 3.84), incorrect answers (avg: 0.61), and noisy or irrelevant content (avg: 1.08). The second row presents answer-level properties: number of correct answers per example (avg: 2.20), wrong answers (avg: 0.86), and number of documents supporting each correct answer (avg: 1.77) and each wrong answer (avg: 0.73).

# B Speculative RAG

Speculative RAG (Wang et al., 2025) first clusters the retrieved documents into $k$ clusters based on their perspectives using an embedding model, InBedder-RoBERTa-large (Peng et al., 2024). From each cluster, it samples one document to form a subset $\delta_j$, repeating this process until unique subsets are created. Each subset $\delta_j$ is fed into a specialist RAG drafter[3] to generate a draft answer $\alpha_j$ and its rationale $\beta_j$. Since our setting may have multiple correct answers, we pass all answers and their rationales to the aggregator to generate the final answer.

# C Astute RAG

Astute RAG (Wang et al., 2024a) aims to address the limitations of imperfect or incomplete retrieval, particularly conflicts between the retrieved context and the LLM's parametric knowledge. The method proceeds in three stages:

**Adaptive Internal Knowledge Generation.** Give a query, the model first generates internal passages (the quantity is determined by the model itself) by prompting itself to recall relevant knowledge without access to any external documents. This step is especially useful when retrieved documents are sparse, noisy, or misleading.

---

[3]As (at time of publication) Wang et al. (2025)'s drafter model is not publicly available, we use different models listed in Table 1 instead.

**Iterative Source-aware Knowledge Consolidation.** Astute RAG combines passages from both internal (LLM-generated) and external (retrieved) sources. Each passage is tagged with its source type (internal or external), which helps the model assess its reliability. The LLM is prompted to identify consistency across passages, detect conflicts, and filter out irrelevant or unreliable information. It reorganizes the input into a smaller set of refined, source-attributed passages. This consolidation step is repeated iteratively to strengthen the context and improve the reliability of the resulting knowledge, especially when dealing with long or conflicting inputs.

**Answer Finalization.** Astute RAG prompts the LLM to generate one answer based on each group of passages. These candidates are then compared, and the LLM selects the most reliable one based on knowledge source, cross-source confirmation, frequency, information thoroughness, and self-assessed confidence.

## D  Computational Efficiency of MADAM-RAG

To improve efficiency, we have incorporated an early stopping criterion in MADAM-RAG: if all agents retain their answers across consecutive rounds, the debate terminates early. In practice, we observe convergence within 1-2 rounds for most examples. To provide a concrete comparison, we calculated the average number of input and output tokens for three representative methods: prompt-based, single agent with self-reflection, and MADAM-RAG.

| Method | Avg Input Tokens | Avg Output Tokens |
|---|---|---|
| Prompt-based | 822.69 | 126.25 |
| Single Agent with Self-reflection | 5033.92 | 809.10 |
| MADAM-RAG (Ours) | 3186.45 | 1547.18 |

Table 2: Average input and output token usage of different methods.

As shown in Table 2, MADAM-RAG requires 3186 input tokens and 1547 output tokens on average—higher than prompt-based RAG (822/126) but comparable to the Single Agent with Self-reflection baseline when taking both input and output tokens into account. This demonstrates that while MADAM-RAG introduces some overhead due to its multi-agent setup, it remains computationally tractable and roughly as efficient as strong single-agent iterative alternatives, while outperforming both zero-shot and single-agent approaches.

## E  Prompts

---
**No RAG**

You are an expert in question answering.
Please respond with the exact answer only. Do not be verbose or provide extra information.
If there are multiple correct answers, please list them all.
Question: {question}
Answer:

---

**Concatenated Prompt**

You are an expert in retrieval question answering.
You will be provided a question with multiple documents. Please answer the question based on the documents.
If there are multiple answers, please provide all possible correct answers and also provide a step-by-step reasoning explanation. If there is no correct answer, please reply 'unknown'.
Please follow the format: "All Correct Answers: []. Explanation: {}"

The following are examples:
Question: In which year was Michael Jordan born?
Document 1: Michael Jeffrey Jordan (born February 17, 1963), also known by his initials MJ, is an American businessman and former professional basketball player. He played 15 seasons in the National Basketball Association (NBA) between 1984 and 2003, winning six NBA championships with the Chicago Bulls. He was integral in popularizing basketball and the NBA around the world in the 1980s and 1990s, becoming a global cultural icon.
Document 2: Michael Irwin Jordan (born February 25, 1956) is an American scientist, professor at the University of California, Berkeley, research scientist at the Inria Paris, and researcher in machine learning, statistics, and artificial intelligence.
Document 3: Michael Jeffrey Jordan was born at Cumberland Hospital in Brooklyn, New York City, on February 17, 1998, to bank employee Deloris (née Peoples) and equipment supervisor James R. Jordan Sr. He has two older brothers, James Jr. and Larry, as well as an older sister named Deloris and a younger sister named Roslyn. Jordan and his siblings were raised Methodist.
Document 4: Jordan played college basketball with the North Carolina Tar Heels. As a freshman, he was a member of the Tar Heels' national championship team in 1982. Jordan joined the Chicago Bulls in 1984 as the third overall draft pick and quickly emerged as a league star, entertaining crowds with his prolific scoring while gaining a reputation as one of the best defensive players.
All Correct Answers: ["1963", "1956"]. Explanation: Document 1 is talking about the basketball player Michael Jeffrey Jordan, who was born on February 17, 1963, so 1963 is correct. Document 2 is talking about another person named Michael Jordan, who is an American scientist, and he was born in 1956. Therefore, the answer 1956 from Document 2 is also correct. Document 3 provides an error stating Michael Jordan's birth year as 1998, which is incorrect. Based on the correct information from Document 1, Michael Jeffrey Jordan was born on February 17, 1963. Document 4 does not provide any useful information.

Question: {question}
{documents list}

## Single Agent with Self-reflection

**Initial Generation:**
You are an expert in retrieval question answering.
You will be provided a question with multiple documents. Please answer the question based on the documents.
If there are multiple answers, please provide all possible correct answers and also provide a step-by-step reasoning explanation. If there is no correct answer, please reply 'unknown'.
Please follow the format: 'All Correct Answers: []. Explanation: {}.'

The following are examples:
Question: In which year was Michael Jordan born?
Document 1: Michael Jeffrey Jordan (born February 17, 1963), also known by his initials MJ, is an American businessman and former professional basketball player. He played 15 seasons in the National Basketball Association (NBA) between 1984 and 2003, winning six NBA championships with the Chicago Bulls. He was integral in popularizing basketball and the NBA around the world in the 1980s and 1990s, becoming a global cultural icon.
Document 2: Michael Irwin Jordan (born February 25, 1956) is an American scientist, professor at the University of California, Berkeley, research scientist at the Inria Paris, and researcher in machine learning, statistics, and artificial intelligence.
Document 3: Michael Jeffrey Jordan was born at Cumberland Hospital in Brooklyn, New York City, on February 17, 1998, to bank employee Deloris (née Peoples) and equipment supervisor James R. Jordan Sr. He has two older brothers, James Jr. and Larry, as well as an older sister named Deloris and a younger sister named Roslyn. Jordan and his siblings were raised Methodist.
Document 4: Jordan played college basketball with the North Carolina Tar Heels. As a freshman, he was a member of the Tar Heels' national championship team in 1982. Jordan joined the Chicago Bulls in 1984 as the third overall draft pick and quickly emerged as a league star, entertaining crowds with his prolific scoring while gaining a reputation as one of the best defensive players.
All Correct Answers: ["1963", "1956"]. Explanation: Document 1 is talking about the basketball player Michael Jeffrey Jordan, who was born on Februray 17, 1963, so 1963 is correct. Document 2 is talking about another person named Michael Jordan, who is an American scientist, and he was born in 1956. Therefore, the answer 1956 from Document 2 is also correct. Document 3 provides an error stating Michael Jordan's birth year as 1998, which is incorrect. Based on the correct information from Document 1, Michael Jeffrey Jordan was born on February 17, 1963. Document 4 does not provide any useful information.

Question: {question}
{context}

- - - - - - - - - - - - - - - - - - - - - - - - - - - - - - - - - - - - - - - - - - - -

**Review Generation:**
You are an expert in retrieval question answering.
You will be provided a question with multiple documents. Please answer the question based on the documents.
If there are multiple answers, please provide all possible correct answers and also provide a step-by-step reasoning explanation. If there is no correct answer, please reply 'unknown'.

Question: {question}
{context}
{answer}

Review your previous answer and find problems with your answer.

- - - - - - - - - - - - - - - - - - - - - - - - - - - - - - - - - - - - - - - - - - - -

**Refine Generation:**
You are an expert in retrieval question answering.
You will be provided a question with multiple documents. Please answer the question based on the documents.
If there are multiple answers, please provide all possible correct answers and also provide a step-by-step reasoning explanation. If there is no correct answer, please reply 'unknown'.

Question: {question}
{context}
{answer}

Review your previous answer and find problems with your answer.
{review}

Based on the problems you found, improve your answer. Please reiterate your answer with the format: 'All Correct Answers: []. Explanation: {}.'

**Speculative RAG**

**Drafter:**
Answer the question based on the documents. Also provide rationale for your response.
Question: {question}
{documents}

Please follow the format: Response: {}. Rationale: {}. You are an expert in question answering.

- - - - - - - - - - - - - - - - - - - - - - - - - - - - - - - - - - - - - - - - - - - - - - - - - -

**Aggregator:**
You are an aggregator reading answers from multiple responses.

If there are multiple answers, please provide all possible correct answers and also provide a step-by-step reasoning explanation. If there is no correct answer, please reply 'unknown'.
Please follow the format: 'All Correct Answers: []. Explanation: {}.'

The following are examples:
Question: In which year was Michael Jordan born?
Responses:
Response 1: Response: 1963. Rationale: The document clearly states that Michael Jeffrey Jordan was born on February 17, 1963.
Response 2: Response: 1956. Rationale: The document states that Michael Irwin Jordan was born on February 25, 1956. However, it's important to note that this document seems to be about a different Michael Jordan, who is an American scientist, not the basketball player. The other agents' responses do not align with the information provided in the document.
Response 3: Response: 1998. Rationale: The According to the document provided, Michael Jeffrey Jordan was born on February 17, 1998.
Response 4: Response: Unknown. Rationale: The provided document focuses on Jordan's college and early professional career, mentioning his college championship in 1982 and his entry into the NBA in 1984, but it does not include information about his birth year.
All Correct Answers: ["1963", "1956"]. Explanation: Response 1 is talking about the basketball player Michael Jeffrey Jordan, who was born on Februray 17, 1963, so 1963 is correct. Response 2 is talking about another person named Michael Jordan, who is an American scientist, and he was born in 1956. Therefore, the answer 1956 from Response 2 is also correct. Response 3 provides an error stating Michael Jordan's birth year as 1998, which is incorrect. Based on the correct information from Response 1, Michael Jeffrey Jordan was born on February 17, 1963. Response 4 does not provide any useful information.

Question: {query}
Responses:
{all drafted responses}

**Astute RAG**

**Adaptive Passage Generation:**
Generate a document that provides accurate and relevant information to answer the given question. If the information is unclear or uncertain, explicitly state 'I don't know' to avoid any hallucinations.

Question: {question} Document:

- - - - - - - - - - - - - - - - - - - - - - - - - - - - - - - - - - - - - - - - - - - -

**Iterative Knowledge Consolidation:**
Task: Consolidate information from both your own memorized documents and externally retrieved documents in response to the given question.

* For documents that provide consistent information, cluster them together and summarize the key details into a single, concise document.
* For documents with conflicting information, separate them into distinct documents, ensuring each captures the unique perspective or data.
* Exclude any information irrelevant to the query.
For each new document created, clearly indicate:
* Whether the source was from memory or an external retrieval.
* The original document numbers for transparency.

Initial Context: {context_init}
Last Context: {context}
Question: {question}
New Context:

- - - - - - - - - - - - - - - - - - - - - - - - - - - - - - - - - - - - - - - - - - - -

**Knowledge Consolidation and Answer Finalization:**
Task: Answer a given question using the consolidated information from both your own memorized documents and externally retrieved documents.

Step 1: Consolidate information
* For documents that provide consistent information, cluster them together and summarize the key details into a single, concise document.
* For documents with conflicting information, separate them into distinct documents, ensuring each captures the unique perspective or data.
* Exclude any information irrelevant to the query.
For each new document created, clearly indicate:
* Whether the source was from memory or an external retrieval.
* The original document numbers for transparency.

Step 2: Propose Answers and Assign Confidence
For each group of documents, propose a possible answer and assign a confidence score based on the credibility and agreement of the information.

Step 3: Select all Correct Answers
After evaluating all groups, select all accurate and well-supported answers.

Initial Context: {context_init}
Consolidated Context: {context} # optional
Question: {question}
Answer:

## MADAM-RAG

**Individual Agent:**
You are an agent reading a document to answer a question.

Question: {question}
Document: {document}

The following responses are from other agents as additional information. # Optional
{history} # Optional

Answer the question based on the document and other agents' responses. Provide your answer and a step-by-step reasoning explanation.
Please follow the format: 'Answer: {}. Explanation: {}.

- - - - - - - - - - - - - - - - - - - - - - - - - - - - - - - - - - - - - - - - -

**Aggregator:**
You are an aggregator reading answers from multiple agents.

If there are multiple answers, please provide all possible correct answers and also provide a step-by-step reasoning explanation. If there is no correct answer, please reply 'unknown'.
Please follow the format: "All Correct Answers: []. Explanation: ."

The following are examples:
Question: In which year was Michael Jordan born?
Agent responses:
Agent 1: Answer: 1963. Explanation: The document clearly states that Michael Jeffrey Jordan was born on February 17, 1963.
Agent 2: Answer: 1956. Explanation: The document states that Michael Irwin Jordan was born on February 25, 1956. However, it's important to note that this document seems to be about a different Michael Jordan, who is an American scientist, not a basketball player. The other agents' responses do not align with the information provided in the document.
Agent 3: Answer: 1998. Explanation: According to the document provided, Michael Jeffrey Jordan was born on February 17, 1998.
Agent 4: Answer: Unknown. Explanation: The provided document focuses on Jordan's college and early professional career, mentioning his college championship in 1982 and his entry into the NBA in 1984, but it does not include information about his birth year.
All Correct Answers: ["1963", "1956"]. Explanation: Agent 1 is talking about the basketball player Michael Jeffrey Jordan, who was born on February 17, 1963, so 1963 is correct. Agent 2 is talking about another person named Michael Jordan, who is an American scientist, and he was born in 1956. Therefore, the answer 1956 from Agent 2 is also correct. Agent 3 provides an error stating Michael Jordan's birth year as 1998, which is incorrect. Based on the correct information from Agent 1, Michael Jeffrey Jordan was born on February 17, 1963. Agent 4 does not provide any useful information.

Question: {question}
Agent responses:
{agent responses list}

