# OpenReview forum: "Retrieval-Augmented Generation with Conflicting Evidence"
_colmweb.org/COLM/2025/Conference — COLM 2025_

### Official Review · Reviewer_YVUj · 2025-05-11

**Rating:** 6
**Confidence:** 3
**Ethics Flag:** 1

**Summary:**

Retrieval-augmented-generation (RAG) often fails when misinformation or conflicting evidences are prestented. Based on the existing benchmark on ambiguous query, this paper proposes a novel dataset consists of multiple factors of conflicting information including noise, misinformation and ambiguity. In experiments, both RAG and recent conflicting information robust RAG (Astute) demonstrates limited performance. To overcome this challenge, the author of the papepr further proposes a multi-agent system: consisting of agent for each supporting document, an aggregation LLM and multi-round of debate, it outperforms existing RAG approaches on three differeent datasets.

**Questions To Authors:**

1. Although the multi-agent detable and aggreagtion performance is examined and justified in the ablation study, I am not fully convinced whether this cannot handled by one LLM with iterations. Can you show or highlight the limtation of a single agent systmem?

2. In section 6.3, the author argued MaDAM-RAG is more robust against number of misinformation. However, I didn't the see the gap becomes bigger when more misinformation is presented.

**Reasons To Accept:**

1. The problem of conflicting information has been a blocker for advancing the accuracy of RAG. This paper proposes a challenging dataset right on the topic and discussed one potential architecture for the problem.

2. The dataset creation involves propoer randomness in each of the conflicting factor, makes the dataset close to the real data distribution.

**Reasons To Reject:**

1. The propose method might fail to scale since each document is assigned to a single agent. If the number of support documents are large (>50), the debate the decision process will inevitablely become slow and expensive.

2. The performance improvements in Table 1 seems pretty mediocore, given the performance improvements over simple RAG on proposed dataset - RAGDocs is 1.8% and 3%, respectively.

---

> ### Author Response · Authors · 2025-06-01
> **Response to Reviewer YVUj**
>
> Thank you for your review and we are glad that you found our dataset RAMDocs to be *“a challenging dataset right on the topic”* and *“close to the real data distribution”*. Please refer to our response to your comments below:
>
> > **Scalability with large numbers of retrieved documents**
>
> In the [**general response**](https://openreview.net/forum?id=z1MHB2m3V9&noteId=6BrPRjAnbl), to provide a concrete comparison of the computational cost of our method, we calculated the average number of input and output tokens for three representative methods, prompt-based, single agent with self-reflection, and MADAM-RAG. As shown in the table, MADAM-RAG requires 3,186 input tokens and 1,547 output tokens on average—higher than prompt-based RAG (822/126) but comparable to the Single Agent with Self-reflection baseline, which consumes over 5,000 input and 800 output tokens per instance. This demonstrates that while MADAM-RAG introduces some overhead due to its multi-agent setup, it remains computationally tractable and roughly as efficient as strong single-agent iterative alternatives, while outperforming both zero-shot and single-agent approaches.
>
> To improve the scalability of the method w.r.t. the number of retrieved documents, future work could additionally include clustering, document pre-filtering, and shared-context compression to reduce the number of agent instances. However, given the focus of our work on developing the RAMDocs benchmark as well as the MADAM-RAG approach, we feel that such practical/engineering contributions are out of the scope of our paper.
>
> > **Performance improvements on RAMDocs**
>
> We would like to clarify that the absolute gains on RAMDocs are on a very challenging testbed combining ambiguity, misinformation, and retrieval noise. As shown in Table 1 and Figure 3 from the original paper, all methods struggle in this setting, and our method consistently yields the highest performance – we argue that this indicates the difficulty of our proposed RAMDocs setting more than anything. Notably, MADAM-RAG outperforms Speculative RAG by 3.8% on RAMDocs and by larger margins on other datasets (e.g., \+13.9% on AmbigDocs), as detailed in the [**general response**](https://openreview.net/forum?id=z1MHB2m3V9&noteId=6BrPRjAnbl). Given the scale and difficulty of these datasets, even modest absolute improvements reflect meaningful gains in robustness and reliability.
>
> > **Question 1: Single LLM with iterative self-reflection**
>
> Thanks for your suggestion, which we address in the [**general response**](https://openreview.net/forum?id=z1MHB2m3V9&noteId=6BrPRjAnbl) by adding self-reflection as a baseline. As shown in the table, MADAM-RAG outperforms the single agent with self-reflection baseline by 17.40% (absolute) on FaithEval, 3.70% on AmbigDocs, and 4.0% on RAMDocs.
> Iterative single-agent methods operate over concatenated or chunked documents and rely on a single LLM to internally manage conflicts. This approach is prone to long-context issues, frequency bias, and missing less frequent but valid answers, especially under ambiguity or evidence imbalance.
>
> > **Question 2: Robustness trend with increasing misinformation**
>
> As shown in Fig. 3 (b) in the original paper, while the performance of all systems declines as misinformation increases, MADAM-RAG yields the highest performance. This degradation is expected since a greater amount of conflicting evidence increases uncertainty around the query, making it more difficult for the model to identify reliable information. These observations are consistent with prior findings \[1\], which show that even strong LLMs can be misled by persuasive misinformation.
>
> \[1\] Rongwu Xu, et al. “The Earth is Flat because...: Investigating LLMs' Belief towards Misinformation via Persuasive Conversation.” ACL 2024

---

> > ### Author Response · Authors · 2025-06-04
> > **Follow-up reminder for Reviewer YVUj**
> >
> > We appreciate the time and effort you’ve put into reviewing our submission. Now that the response period is underway, we want to follow up and see if our response has addressed your comments. If our additional experiments and clarifications have resolved your comments, we would appreciate it if you could revisit your score accordingly. We’re happy to continue discussing any points in the remaining days of the rebuttal period.

---

> > > ### Comment · Reviewer_YVUj · 2025-06-09
> > >
> > > Thanks for the response. It addresses most of my concerns and I am willing to raise my score. My remaining question is why the RAMDocs multi-agent token consumption is less than the single-agent baseline. what's the hyper-parameter of the self-reflection baseline?

---

> > > > ### Author Response · Authors · 2025-06-10
> > > > **Response to YVUj regarding token consumption**
> > > >
> > > > Thank you for your follow-up and for considering raising your score. We are glad that most of your concerns have been addressed.
> > > >
> > > > Regarding your question about token consumption: the single-agent with self-reflection baseline requires significantly more input tokens because it performs two full iterations over the entire concatenated context. Each iteration includes a long prompt, the model’s previous answer, and its self-critique, resulting in a substantial accumulation of both input and output tokens across turns. In contrast, MADAM-RAG distributes documents across agents, each of which processes only a single document rather than the full retrieved set. Although MADAM-RAG involves multiple rounds of debate, the input seen by each agent is much smaller, and we implement an early stopping criterion that typically decreases the number of rounds to 2/3 in practice, helping to reduce unnecessary computation. As for the self-reflection baseline, we use only one hyperparameter: the number of reflection rounds, which is set to 2.

---

### Official Review · Reviewer_MWio · 2025-05-12

**Rating:** 6
**Confidence:** 5
**Ethics Flag:** 1

**Summary:**

Summary of strong points:

1. Novel Framework to handle conflicting evidence scenario in RAG: a multiagent debate setup (MADAM-RAG) to resolve conflicting evidence, which is an innovative way to combine the strengths of several agents and allow refinement over multiple rounds. The iterative discussion and aggregation mechanism can improve the robustness of retrieval-augmented responses, especially in scenarios with ambiguous queries.

2. Timely topic: focuses on timely issue given the popularity of RAG system in real world use cases.

3. Superior performance: Empirical results demonstrate that the multiagent debate approach achieves improved performance under conditions of conflicting or noisy evidence.

Summary of weak points:
1. Methodological Details of agentic system: The paper needs could be more explicit about the debate mechanism, details on how multiple conversation rounds are organized and how each agent revises their answer would help clarify the process. The design of the aggregator—how it collects and biases contributions from various agents.

2. Most important, additional computational overhead : The multiagent debate approach may introduce increased computational costs.

3. Not much novelity on proposed benchmark dataset (RAMDOCS)

4. Very few baseline models.

**Reasons To Accept:**

1. Originality via conceptual Integration: Multiagent system of discussion and aggregation is getting popular in writing use case (like Stanford STORM). Bringing this idea to RAG under conflicting evidence case is innovative and effective as seen from the reported results.

2. Strong and consistent performance of the proposed method over baselines

**Reasons To Reject:**

1. Single agent RAG systems have low computational overhead. Incorporating multiagent system in retrieval would significantly increase the computational cost, thereby hindering  the applicability of this method in real-world use cases. Further discussion on inference overhead, latency, or scalability compared to single-agent systems is needed.

2. Need more comprehensive comparison across different models and metrics especially with long context LLMs.

---

> ### Author Response · Authors · 2025-06-01
> **Response to Reviewer MWio**
>
> Thank you for your review and we are glad that you appreciate that MADAM-RAG is a *“novel framework to handle conflicting evidence scenario in RAG*” and it has “*superior performance*”. Please find the detailed response to your individual comments below:
>
> > **Methodological details of agentic system**
>
> As mentioned in Section 4, each agent is assigned a unique document and generates an intermediate response independently in the first round. In subsequent rounds, each agent receives the aggregator’s summary of all responses from the prior round and may revise its answer accordingly. This allows agents to incorporate cross-document signals while maintaining their independence.
>
> The aggregator takes the full set of agent responses and generates a unified summary answer and an explanatory rationale. Its prompt explicitly encourages attention to consistency, evidence traceability, and disambiguation. In ambiguous settings, it may preserve multiple valid answers, while in misinformation settings, it aims to discard unsupported claims. We will incorporate the details in the final paper.
>
> > **Computational efficiency**
>
> In the [**general response**](https://openreview.net/forum?id=z1MHB2m3V9&noteId=6BrPRjAnbl), we provide a concrete comparison of computational efficiency. Here, we calculated the average number of input and output tokens for three representative methods, prompt-based, single agent with self-reflection, and MADAM-RAG. As shown in the table, MADAM-RAG requires 3,186 input tokens and 1,547 output tokens on average—higher than prompt-based RAG (822/126) but comparable to the Single Agent with Self-reflection baseline, which consumes over 5,000 input and 800 output tokens per instance. This demonstrates that while MADAM-RAG introduces some overhead due to its multi-agent setup, it remains computationally tractable and roughly as efficient as strong single-agent iterative alternatives, while outperforming both zero-shot and single-agent approaches.
>
> > **Novelty of the proposed benchmark**
>
> While RAMDocs builds on AmbigDocs, it introduces key extensions to simulate real-world retrieval more faithfully. Specifically, it combines multiple sources of conflict—ambiguity, misinformation, and noise—in a single query instance. It also varies supporting document distributions, creating realistic imbalances that challenge majority-biased systems. Our results (Section 5.2, Figure 3\) show that RAMDocs is substantially more difficult than existing benchmarks. We believe this multifactor composition and the diagnostic nature of the dataset represent a meaningful step forward, as highlighted by Reviewer YVUj (*“this paper proposes a novel dataset” which addresses “a blocker for advancing the accuracy of RAG”*) and by Reviewer Zrjy, who mentions that our paper *“presents a realistic benchmark that effectively captures key challenges faced by practical RAG systems”.*
>
> > **Additional baselines**
>
> Thank you for this suggestion, which we have addressed in the [**general response**](https://openreview.net/forum?id=z1MHB2m3V9&noteId=6BrPRjAnbl), We compare MADAM-RAG against three more baselines: Single Agent with Self-reflection, Self-RAG, and Speculative RAG (details in the general response). As shown in the table, MADAM-RAG consistently outperforms three additional baselines across all three datasets, which demonstrates the effectiveness of our method.
>
> > **Long context LLMs**
>
> We would like to clarify that our results already include long-context LLMs. Specifically, we have included results with GPT-4o-mini, a long-context-capable model \[1\], as shown in Table 1\. The results show that MADAM-RAG still achieves the best performance across all three datasets.
>
> \[1\] https://openai.com/index/gpt-4o-mini-advancing-cost-efficient-intelligence

---

> > ### Author Response · Authors · 2025-06-04
> > **Follow-up reminder for Reviewer MWio**
> >
> > We appreciate the time and effort you’ve put into reviewing our submission. Now that the response period is underway, we want to follow up and see if our response has addressed your comments. If our additional experiments and clarifications have resolved your comments, we would appreciate it if you could revisit your score accordingly. We’re happy to continue discussing any points in the remaining days of the rebuttal period.

---

> > ### Comment · Reviewer_MWio · 2025-06-07
> >
> > Thanks to the authors for the response. My concern on computational efficiency is cleared now. Its definitely more expensive than standard RAG systems but efficient than many agentic system.

---

> > > ### Author Response · Authors · 2025-06-07
> > > **Response to follow-up by Reviewer MWio**
> > >
> > > Thanks for the continued engagement -- we are glad that we have addressed your comment on computational efficiency. We hope that our **additional details on the design of MADAM-RAG, clarifications on novelty and long-context LLMs, and positive results with strong improvements over three additional baselines** (Single Agent w/ Self-Reflection [1,2], Self-RAG [3], and Speculative RAG [4])  address your other questions/comments.
> > >
> > > We hope that these results together will allow you to revisit your score, and are happy to provide any additional clarification in the few days of the rebuttal that remain.
> > >
> > > [1] https://arxiv.org/abs/2303.17651
> > >
> > > [2] https://arxiv.org/abs/2310.01798
> > >
> > > [3] https://arxiv.org/abs/2310.11511
> > >
> > > [4] https://arxiv.org/abs/2407.08223

---

### Official Review · Reviewer_Zrjy · 2025-05-18

**Rating:** 6
**Confidence:** 4
**Ethics Flag:** 1

**Summary:**

The paper focuses on retrieval augmented generation, which is a important task of LLMs. The main challenge is to handle potential hallucinations from the LLM responses and the ambiguous queries from the user. This paper well studies both aspects by proposing a new benchmark and a targeted method.

The benchmark is established based on another existing one which studied the ambiguous user query (AmbigDocs). The new benchmark augments the original one by adding supporting documents. In this new benchmark, the authors control the ambiguity of the query by sampling 1-3 correct answers for each query and add 1-3 documents for each possible answer. These practices are to make the dataset closer to the realistic RAG scenario.

The method is based on multi-agent system. Each agent is generating answers based on a specific piece of document and their outputs are aggregated by a special module called aggregator to summarize their difference and correlation. Such processing is conducted iteratively so in the later iterations, the individual agent would learn from the other agents' previous answer and polish their own answer.

**Questions To Authors:**

1. Figure 2 suggests that agents in the system process aggregated results in a specific sequence. Is this observation accurate? If so, what methodology determines this processing order?

2. Beyond answer accuracy metrics, I'm interested in the system's computational efficiency. Could you provide information on the total token consumption per question across the entire system?

**Reasons To Accept:**

1. The paper presents a realistic benchmark that effectively captures key challenges faced by practical RAG systems: ambiguous user queries and variable numbers of supporting documents.

2. The proposed system utilizes a multi-agent approach to effectively manage conflicts and correlations between retrieved documents. Its iterative design enables individual agents to learn from one another, enhancing overall performance.

**Reasons To Reject:**

1. While the temporary answer modules are labeled as "agents," they essentially perform the same function of generating responses based on individual documents. Using the term "agent" or describing the system as "multi-agent" may be misleading, as these terms carry specific connotations in the current literature that don't align with the implementation described.

2. The comparison baselines appear insufficient. Although some related methods may have proprietary implementations, additional open-source alternatives could strengthen the evaluation. Consider including other relevant baselines such as Self-RAG (already in the citations).

3. The paper lacks comprehensive citation coverage. Many previous works have explored RAG systems that first analyze individual retrieval results or subsets before performing aggregation or ranking. These relevant approaches are not adequately cited or compared against. For example, Speculative RAG* employs a similar methodology of distributing documents and then aggregating answers. More thorough discussion in the related works section would strengthen the paper's comprehensiveness and proper acknowledgment of prior contributions.

- Asai, Akari, et al. "Self-rag: Learning to retrieve, generate, and critique through self-reflection." The Twelfth International Conference on Learning Representations. 2024.
- Wang, Zilong, et al. "Speculative rag: Enhancing retrieval augmented generation through drafting." The Thirteenth International Conference on Learning Representations. 2025.

---

> ### Author Response · Authors · 2025-06-01
> **Response to Reviewer Zrjy**
>
> Thanks for your review and for highlighting the fact that our paper “*presents a realistic benchmark that effectively captures key challenges faced by practical RAG systems*”. We have conducted additional experiments to include three more baselines and show that MADAM-RAG outperforms them, as detailed in the [**general response**](https://openreview.net/forum?id=z1MHB2m3V9&noteId=6BrPRjAnbl). We have sought to address your remaining comments below:
>
> > **Use of the term “multi-agent”**
>
> We agree that the term “multi-agent” has been used in varying ways across different papers. In our context, each instance of the LLM, instantiated with isolated input and persistent dialogue memory, is referred to as an agent; this follows the discussion presented in \[1\]. Each agent in our method is independently instantiated with a single document and engages in multiple rounds of discussion, updating responses based on prior global summaries. While some work has used the term “multi-agent” to refer to modular pipelines of agents performing different tasks \[2\], our use of multi-agent is consistent with a large body of work on multi-agent debate/discussion, where agents perform the same task \[3,4,5\]. We will clarify this point in the final version.
>
> \[1\] [https://gist.github.com/yoavg/9142e5d974ab916462e8ec080407365b](https://gist.github.com/yoavg/9142e5d974ab916462e8ec080407365b)
> \[2\] [https://arxiv.org/pdf/2308.08155](https://arxiv.org/pdf/2308.08155)
> \[3\] [https://arxiv.org/abs/2309.13007](https://arxiv.org/abs/2309.13007)
> \[4\] [https://arxiv.org/abs/2305.14325](https://arxiv.org/abs/2305.14325)
> \[5\] [https://arxiv.org/abs/2308.07201](https://arxiv.org/abs/2308.07201)
>
> > **Comparison with Self-RAG.**
>
> Based on this suggestion, in the [**general response**](https://openreview.net/forum?id=z1MHB2m3V9&noteId=6BrPRjAnbl), we compare MADAM-RAG with Self-RAG on Llama3.3-70B-Inst as a representative model. As shown in the table, MADAM-RAG consistently outperforms Self-RAG across all three datasets. The gains are particularly notable on FaithEval and RAMDocs, where misinformation needs to be identified and removed; for instance, MADAM-RAG outperforms Self-RAG by 30.70%(absolute) on FaithEval and 7.60% on RAMDocs. We thank you for your suggestion and will include this result using an extra page in the final version.
>
> > **Related Work Coverage and comparison with Speculative RAG.**
>
> We have cited some of these RAG works, such as Self-RAG (Line 122), Astute RAG (Line 123), and Main-RAG (Line 126). While existing RAG systems (e.g., Speculative RAG) also have parallel document-level processing followed by aggregation, they fundamentally differ from MADAM-RAG in that they do not support multi-turn debate among agents. Our framework enables each document-conditioned agent to revise its answer iteratively based on the summarized responses of others, a mechanism that promotes conflict resolution, minority answer preservation, and misinformation suppression in a more dynamic and interactive way. In the [**general response**](https://openreview.net/forum?id=z1MHB2m3V9&noteId=6BrPRjAnbl), we show empirically that MADAM-RAG outperforms Speculative RAG by 1.3% (absolute) on FaithEval, 13.9% on AmbigDocs, and 3.8% on RAMDocs, which demonstrates the effectiveness and robustness of our multi-agent debate method. We appreciate your suggestion and will add these additional results in the final version.
>
> > **Question 1: agent processing order**
>
> Thank you for pointing this out. We do not process aggregated results in a specific sequence. To mitigate the position bias, we shuffle the responses from agents before passing them to the aggregator. We will clarify this point in the final version.
>
> > **Question 2: token-level computational efficiency**
>
> In the [**general response**](https://openreview.net/forum?id=z1MHB2m3V9&noteId=6BrPRjAnbl), we provide a concrete comparison of computational efficiency. Here, we calculated the average number of input and output tokens for three representative methods, prompt-based, single agent with self-reflection, and MADAM-RAG (see the second point in the general table). As shown in the table, MADAM-RAG requires 3,186 input tokens and 1,547 output tokens on average—higher than prompt-based RAG (822/126) but comparable to the Single Agent with Self-reflection baseline, which consumes over 5,000 input and 800 output tokens per instance. This demonstrates that while MADAM-RAG introduces some overhead due to its multi-agent setup, it remains computationally tractable and roughly as efficient as strong single-agent iterative alternatives, while outperforming both zero-shot and single-agent approaches.

---

> > ### Author Response · Authors · 2025-06-04
> > **Follow-up reminder for Reviewer Zrjy**
> >
> > We appreciate the time and effort you’ve put into reviewing our submission. Now that the response period is underway, we want to follow up and see if our response has addressed your comments. If our additional experiments and clarifications have resolved your comments, we would appreciate it if you could revisit your score accordingly. We’re happy to continue discussing any points in the remaining days of the rebuttal period.

---

> > > ### Comment · Reviewer_Zrjy · 2025-06-09
> > >
> > > Thank you for the thorough rebuttal. I’ve read it and updated my score accordingly.

---

> > > > ### Author Response · Authors · 2025-06-10
> > > >
> > > > Thank you for raising your score -- we appreciate the discussion and will incorporate these updates into our final paper.

---

### Official Review · Reviewer_kbmP · 2025-05-20

**Rating:** 7
**Confidence:** 4
**Ethics Flag:** 1

**Summary:**

In this paper, the authors propose an RAG dataset that considers query ambiguity and misinformation in documents. They further propose Madam-RAG, a multi-agent approach that jointly handles diverse sources of conflict using multiple LLM-based agents. Results verified the effectiveness of the proposed method.


Pros:
- RAG is a trending research topic in the area of large language models. The problem discussed in the paper is important and interesting.
- Using multi-agents to handle noises and conflicts in RAG is interesting
- The proposed method is reasonable and relatively solid. The proposed dataset is useful to the research community.
- The paper is well written and easy to follow.

Cons:
- The motivation and justification for why using multiple agents could work well for this task should be better introduced and explained. Why cannot a single agent or an LLM with self-reflection solve this problem? The current introduction to this is not convincing.
- What is the relationship between "conflict" and "ambiguity, misinformation, and noise"?

**Reasons To Accept:**

- RAG is a trending research topic in the area of large language models. The problem discussed in the paper is important and interesting.
- Using multi-agents to handle noises and conflicts in RAG is interesting
- The proposed method is reasonable and relatively solid. The proposed dataset is useful to the research community.
- The paper is well written and easy to follow.

**Reasons To Reject:**

- The motivation and justification for why using multiple agents could work well for this task should be better introduced and explained. Why cannot a single agent or an LLM with self-reflection solve this problem? The current introduction to this is not convincing.
- What is the relationship between "conflict" and "ambiguity, misinformation, and noise"?

---

> ### Author Response · Authors · 2025-06-01
> **Response to Reviewer kbmP**
>
> Thank you for your review and for appreciating our “_reasonable and relatively solid_” approach and the utility of our proposed dataset. We have conducted additional experiments to include three more baselines and show that MADAM-RAG outperforms them, as detailed in the [**general response**](https://openreview.net/forum?id=z1MHB2m3V9&noteId=6BrPRjAnbl). Please find responses to the remaining comments/questions below.
>
> > **Multi-agent vs. single agent with self-reflection**
>
> Based on this suggestion, we have additionally implemented a single agent with self-reflection and evaluated it on all three datasets, with the full results detailed in the [**general response**](https://openreview.net/forum?id=z1MHB2m3V9&noteId=6BrPRjAnbl).  MADAM-RAG outperforms the single agent with self-reflection baseline by 17.40% (absolute) on FaithEval, 3.70% on AmbigDocs, and 4.0% on RAMDocs. We will incorporate this baseline into the final version.
>
> These results show some of the shortcomings of single-agent systems: while single-agent approaches like self-reflection\[1,2\] can identify internal inconsistencies, they still process concatenated input and are therefore vulnerable to context length limits, frequency bias, and order effects\[3,4\]. In our setting, as shown in Fig. 3, processing all documents jointly may obscure low-frequency but valid evidence, especially when misinformation or noise dominates the retrieved set. Our multi-agent setup encourages each document to be independently summarized and defended, allowing underrepresented but correct views to be preserved and strengthened through debate.
>
> \[1\] Aman Madaan, et al. "Self-refine: Iterative refinement with self-feedback." NeurIPS 2023
> \[2\] Jie Huang, et al. "Large Language Models Cannot Self-Correct Reasoning Yet." ICLR 2024
> \[3\] Nelson F. Liu, et al. “Lost in the Middle: How Language Models Use Long Contexts.” TACL 2023
> \[4\] Tony Z. Zhao, et al. “Calibrate Before Use: Improving Few-Shot Performance of Language Models.” ICML 2021
>
> > **Relationship between "conflict" and "ambiguity, misinformation, and noise"**
>
> We frame the conflict in our paper (Line 47-52) as inter-context conflict which is the disagreement among retrieved contexts (documents). These conflicts arise from different causes:
>
> * Following \[5\], we treat ambiguity as cases where multiple retrieved documents offer different but valid answers due to underspecified queries.
> * Following \[6\] misinformation is defined as documents containing factually incorrect information.
> * We define noise as documents are irrelevant or only weakly related to the query.
>
> We will clarify these distinctions further in the final version.
>
> \[5\] Yoonsang Lee, et al. "AmbigDocs: Reasoning across Documents on Different Entities under the Same Name." COLM 2024
>
> \[6\] Yifei Ming, et al. "FaithEval: Can Your Language Model Stay Faithful to Context, Even If 'The Moon is Made of Marshmallows'." ICLR 2025

---

> > ### Author Response · Authors · 2025-06-04
> > **Follow-up reminder for Reviewer kbmP**
> >
> > We appreciate the time and effort you’ve put into reviewing our submission. Now that the response period is underway, we want to follow up and see if our response has addressed your comments. We welcome any further discussion on your comments and are happy to clarify additional points during the remaining days of the rebuttal period.

---

> ### Comment · Reviewer_kbmP · 2025-06-06
>
> Thank the authors for the response.
> How is your single-agent-based method implemented? Note that a single agent doesn't mean that it must process all documents jointly, right? It can be a single agent that handles each task separately, just like a person can handle different tasks independently.

---

> > ### Author Response · Authors · 2025-06-06
> > **Follow-up Response to Reviewer kbmP**
> >
> > Thanks for your follow-up and continued engagement. In our implementation of the single agent with self-reflection baseline, we follow the widely-used formulation in prior work [1,2], where the model processes the full concatenated context and then revises its answer through multiple rounds of self-critique. However, we also have results that we believe follow the single-agent setting you describe in your comment “Note that a single agent doesn’t mean that it must process all documents jointly, right? It can be a single agent that handles each task separately”. Specifically, in Table 2 we show an ablation across rounds and with/without the aggregator agent. In round 1, the agents have not yet discussed their answers with each other, and so this can be thought of as a “single agent” processing each document sequentially (with or without a subsequent aggregator agent).
> >
> > Based on your suggestion, we have expanded this analysis to also include RAMDocs and show accuracy, precision, recall, and F1, with the results for round 1 summarized below. In both cases, we generally see lower accuracy and precision numbers, especially for the true “single-agent” setting (no aggregator) where the accuracy and precision are substantially lower.
> >
> > **FaithEval Table**:
> > | Method | Acc. | P | R |  F1 |
> > |-|:-:|:-:|:-:|:-:|
> > | w/ aggregator | 37.80 | 55.10 | 72.80 | 61.00 |
> > | w/o aggregator | 8.30 | 44.30 | 76.70 | 55.10 |
> >
> > **RAMDocs Table**:
> > | Method | Acc. | P | R |  F1 |
> > |-|:-:|:-:|:-:|:-:|
> > | w/ aggregator | 37.80 | 67.40 | 74.00 | 68.60 |
> > | w/o aggregator | 30.00 | 50.30 | 79.70 | 59.80 |
> >
> > [1] Aman Madaan, et al. "Self-refine: Iterative refinement with self-feedback." NeurIPS 2023
> >
> > [2] Jie Huang, et al. "Large Language Models Cannot Self-Correct Reasoning Yet." ICLR 2024

---

### Author Response · Authors · 2025-06-01
**General Response: Additional Baseline and Computational Cost**

We would like to thank all the reviewers for their thorough reviews and comments. Multiple reviewers emphasized the importance and timeliness of the problem setting, noting that our benchmark, RAMDocs, addresses a key “*blocker for advancing the accuracy of RAG*” (YVUj) is “*useful to the research community*” (kbmP), and “*presents a realistic benchmark that effectively captures key challenges faced by practical RAG systems*” (Zrjy). Reviewers also praised the multi-agent debate framework as “*innovative and effective*” (MWio) and highlighted its *effectiveness* (kbmP). Here, we summarize changes made to address shared comments between reviewers, with responses to each reviewer’s specific points provided in their respective responses.

> **Added three baselines (Single Agent with Self-reflection, Self-RAG, Speculative RAG)**

To underscore the need for a multi-agent approach, we compare MADAM-RAG against two single-agent baselines and one additional baseline that uses drafter and verifier agents:

* **Single Agent with Self-reflection**: Following \[1,2\], we apply a three-step prompting strategy for self-reflection: 1\) prompt the model to perform an initial generation; 2\) prompt the model to review its previous generation and produce feedback; 3\) prompt the model to answer the original question again with the feedback. We conduct two rounds of self-reflection.
* **Self-RAG** \[3\]: Self-RAG instruction-tunes an LM (Llama2-13B) to generate special self-reflection tags. These tags guide the LM to dynamically retrieve documents when necessary, critique the retrieved documents’ relevance before generating responses. The Self-RAG model processes multiple retrieved documents in parallel and generates separate responses for each document. Since our setting may have multiple correct answers, we pass all responses with their critique scores to the aggregator to generate the final answer, instead of following the original method to select the response with the highest critique score as the final answer.
* **Speculative RAG** \[4\]: first clusters the retrieved documents into $k$ clusters based on their perspectives using an embedding model, InBedder-RoBERTa-large \[5\]. From each cluster, it samples one document to form a subset $\\delta\_j$, repeating this process until $m$ unique subsets are created. Each subset $\\delta\_j$ is fed into a specialist RAG drafter (since the authors haven’t released their drafter model yet, we use Llama3.3-70B-Inst) to generate a draft answer $\\alpha\_j$ and its rationale $\\beta\_j$. Since our setting may have multiple correct answers, we pass all answers and their rationales to the aggregator to generate the final answer.

| Model | Method | FaithEval | AmbigDocs | RAMDocs |
|-|-|:-:|:-:|:-:|
| SelfRAG-Llama2-13B | Self-RAG | 12.40 | 55.00 | 26.80 |
| Llama3.3-70B-Inst | Prompt-based | 27.30 | 54.20 | 32.60 |
| Llama3.3-70B-Inst | Single Agent with Self-reflection | 25.70 | 54.50 | 30.40 |
| Llama3.3-70B-Inst | Speculative RAG | 41.80 | 44.30 | 30.60 |
| Llama3.3-70B-Inst | MADAM-RAG (Ours) | **43.10** | **58.20** | **34.40** |

As shown in the above tables, MADAM-RAG consistently outperforms these strong baselines across all three datasets. These results reinforce that MADAM-RAG’s structured multi-agent framework provides more reliable conflict resolution than single-agent reflection or clustering-based draft selection.

\[1\] https://arxiv.org/abs/2303.17651

\[2\] https://arxiv.org/abs/2310.01798

\[3\] https://arxiv.org/abs/2310.11511

\[4\] https://arxiv.org/abs/2407.08223

\[5\] https://arxiv.org/abs/2402.09642

> **Computational efficiency of MADAM-RAG**

To improve efficiency, we have incorporated an early stopping criterion in MADAM-RAG (Line 261-266): if all agents retain their answers across consecutive rounds, the debate terminates early. In practice, we observe convergence within 1-2 rounds for most examples. To provide a concrete comparison, we calculated the average number of input and output tokens for three representative methods: prompt-based, single agent with self-reflection, and MADAM-RAG.

| Method | Avg Input Tokens | Avg Output Tokens |
|-|:-:|:-:|
| Prompt-based | 822.69 | 126.25 |
| Single Agent with Self-reflection | 5033.92 | 809.10 |
| MADAM-RAG (Ours) | 3186.45 | 1547.18 |

As shown in the table above, MADAM-RAG requires 3186 input tokens and 1547 output tokens on average—higher than prompt-based RAG (822/126) but comparable to the Single Agent with Self-reflection baseline when taking both input and output tokens into account. This demonstrates that while MADAM-RAG introduces some overhead due to its multi-agent setup, it remains computationally tractable and roughly as efficient as strong single-agent iterative alternatives, while outperforming both zero-shot and single-agent approaches.

---

### Comment · Area_Chair_ZbUc · 2025-06-08
**Author-Reviewer Discussions**

Dear reviewers,

Thanks for your review! The discussion period is ending soon, and it'd be greatly appreciated if you could respond to the authors and update your review if necessary.

Reviewers Zrjy/YVUj: Looks like you haven't acknowledged the author's rebuttal. Could you please do so at your earliest convenience?

Thank you!
AC

---

### Decision · Program_Chairs · 2025-07-08

**Decision:**

Accept

**Comment:**

This paper studies an important problem in RAG systems: effectively handling ambiguous user queries and conflicting information from multiple retrieval sources, including misinformation and noise. The authors propose two main contributions: (1) RAMDocs, a novel dataset designed to simulate complex scenarios with conflicting evidence, and (2) MaDAM-RAG, a multi-agent debate-based approach to jointly manage these diverse sources of conflict. The reviewers generally agree on the importance and relevance of the problem and appreciate the novelty of the multi-agent approach. The concerns regarding methodological justification, computational overhead, and lack of several baseline comparisons were partially addressed during the rebuttal. The AC believes that the paper warrants acceptance, and also suggests the authors include discussions of related work/baselines with similar motivation/applications, such as the following RAG papers:
* RankRAG: Unifying Context Ranking with Retrieval-Augmented Generation in LLMs (NeurIPS'24)
* InstructRAG: Instructing Retrieval-Augmented Generation via Self-Synthesized Rationales (ICLR'25)